# Laser Ablation ICP-MS of Size-Segregated Atmospheric Particles Collected with a MOUDI Cascade Impactor: A Proof of Concept

Marin S. Robinson<sup>1</sup>, Irena Grgić<sup>2</sup>, Vid S. Šelih<sup>2</sup>, Martin Šala<sup>2</sup>, Marsha Bitsui<sup>1</sup>, Johannes T. van Elteren<sup>2</sup>

5

<sup>1</sup>Department of Chemistry and Biochemistry, Northern Arizona University, P.O. Box 5698, Flagstaff, AZ, U.S. 86011 <sup>2</sup>Department of Analytical Chemistry, National Institute of Chemistry, Slovenia, Hajdrihova 19, SI-1000 Ljubljana, Slovenia

Correspondence to: Johannes T. van Elteren (elteren@ki.si)

# 10

**Abstract.** A widely used instrument for collecting size-segregated particles is the micro-orifice uniform deposit impactor (MOUDI). In this work, a 10-stage MOUDI (cutpoint dia. 10  $\mu$ m to 56 nm) was used to collect samples in Ljubljana, Slovenia and Martinska, Croatia. Filters, collected with and without rotation, were cut in half and analysed for nine elements (As, Cu, Fe, Ni, Mn, Pb, Sb, V, Zn) using laser ablation ICP-MS. Elemental image maps (created with ImageJ) were converted to concentrations using NIST SRM 2783. Statistical analysis of the elemental maps indicated that for submicron particles (stages 6–10), ablating 10 % of the filter (0.5 cm<sup>2</sup>, 20 min ablation time) was sufficient to give values in good agreement (±10 %) to analysis of larger parts of the filter and with good precision (*RSE* < 1 %). Excellent sensitivity was also observed (e.g., 20 ± 0.2 pg m<sup>-3</sup> V). The novel use of LA ICP-MS, together with image mapping, provided a fast and sensitive method for elemental analysis of size-segregated MOUDI filters, particularly for submicron particles.

#### **1** Introduction

Inhalation of particle-bound metals in atmospheric particulate can negatively impact human health (Chen and Lippmann, 2009). Particle-bound Fe, Ni, and V can lead to oxidative stress, pulmonary inflammation, cardiac effects, and cardiovascular and respiratory illnesses (Aust et al., 2002; Bell et al., 2009; Campen et al., 2002). Particle size is also a factor. Submicron particles pose the greatest health risks (Davidson et al., 2005), and particle-bound metals from anthropogenic sources (e.g., fossil fuel combustion and vehicle emissions) commonly partition to these smaller sizes (<1 µm) (Fang and Huang, 2011). A well-known instrument for the collection and speciation of size-segregated particles is the multi-orifice uniform deposit impactor (MOUDI) (Allen et al., 2001; Herner et al., 2006; Ntziachristos et al., 2007; Pekney

- et al., 2006; Singh et al., 2002). Particles are sorted by size into stages using cascade impaction and deposited on filters. Models with various flow rates (2-130 L min<sup>-1</sup>) and stages (cutpoint dia. between 0.01 and 18 μm) are available, as well as rotating and non-rotating options (Marple, 1991; Marple et al., 2014). To determine elemental concentrations, filters are typically extracted with acid and analyzed by ICP-MS (Canepari et al., 2008; Herner et al., 2006; Li et al., 2012; Ntziachristos et al., 2007; Pekney et al., 2006) or ICP-AES (Fang and Huang, 2011), whereas in some cases direct analysis
- of the filters was performed by X-ray fluorescence spectrometry<sup>11</sup> or proton-induced X-ray emission (Brüggeman et al.,

2009). Detection limits using these methods are challenging due to low particle mass and contamination risks; after corrections have been made for blanks, elemental concentrations are often below detection limits (Pekney et al., 2006).

A promising alternative to acid extraction involves laser ablation ICP-MS (LA ICP-MS) (Aubriet and Carré, 2010). This method samples the filters via pulsed laser ablation, effectively removing the particles, thereby eliminating the need for harsh chemicals, offering faster sample preparation times, reducing contamination, and increasing sensitivity. Initial efforts using LA ICP-MS have been promising (Gligorovski et al., 2008; Triglav et al., 2010), although problems associated with spatial inhomogeneity (Brown et al., 2009), matrix-matched standards (Chin et al., 1999; Tanaka et al., 1998), and laser instabilities have been reported. Hsieh et al. (2011) optimized the LA-ICP-MS parameters suitable for analysis of nanometer-and submicrometer-sized airborne particulate matter sampled by an electrical low-pressure impactor. In this work, we addressed these problems by using a MOUDI for particle collection and a highly stable excimer laser (193 nm ArF\*) for particle ablation. To our knowledge, we are the first to analyze MOUDI filters by LA ICP-MS. Filters collected both with and without rotation were analyzed. Concentrations of nine elements from ten MOUDI stages are reported with specific

attention given to submicron particles (stages 6–10).

#### 2 Materials and methods

# 15 2.1 Air Sampling

A 10-stage, micro-orifice uniform-deposit impactor (MOUDI, Model 110R with rotator, Applied Physics, Inc.) was used to collect four air samples where stages 1–10 correspond to 3, 10, 10, 20, 40, 80, 900, 900, 2000, 2000 nozzles and 50% cutpoint dia. of 10, 5.6, 3.2, 1.8, 1.0, 0.56, 0.32, 0.18, 0.10, and 0.0156 μm, respectively. Samples 1 (24 h, March 3 to 4, 2015) and 2 (72 h, Mar 31 to April 3, 2015) were collected in a residential area of Ljubljana, Slovenia. Samples 3 (May 6 to 7, 2015) and 4 (May 8 to 9, 2015) were collected at Martinska station, Croatia, an estuary on the Eastern Adriatic coast. Particles were collected on PTFE filters (Whatman, 46.2 mm, deposit area 4.91 cm<sup>2</sup>). Samples 1, 2, and 4 were collected with rotation (depositing particles in concentric circles); sample 3 was collected without rotation (depositing particles in mounds or spots). The nominal flow rate was 30 L min<sup>-1</sup>.

25

#### 2.2 Laser ablation ICP-MS

A quadrupole ICP-MS (Agilent Technologies 7900, Palo Alto, USA) interfaced with a laser ablation system (193 nm ArF\* excimer, Analyte G2, Teledyne Photon Machines Inc.) was used to analyze nine nuclides (<sup>75</sup>As, <sup>63</sup>Cu, <sup>57</sup>Fe, <sup>55</sup>Mn, <sup>60</sup>Ni, <sup>208</sup>Pb,

30 <sup>121</sup>Sb, <sup>51</sup>V, <sup>66</sup>Zn). (See optimized parameters in Table 1.) Ablation took place in a HelEx 2-volume cell applying a laser beam size of 150 μm (square mask), a scanning speed of 300 μm s<sup>-1</sup>, a repetition rate of 10 Hz, and a fluence of 1.21 J cm<sup>-2</sup>. Ablated materials were transported from the ablation cell to the plasma with helium; argon was added as the make-up gas

before the ICP torch. Ions formed in the plasma were extracted and separated by their mass-to-charge (m/z) ratios. The mass spectrometer, in time-resolved analysis mode, measured one point per mass for the nine selected masses. The detection limit for each element was determined as  $3 \times SD$  of seven clean blanks (Table 2).

# 5 2.3 Filter preparation and ablation

All 10 filters (stages 1–10) were ablated in samples 1 and 2 (Ljubljana, rotation), eight filters (stages 3–10) were ablated in sample 3 (Martinska, non-rotation), and four filters (samples 6–9) were ablated in sample 4 (Martinksa, rotation). Filters were cut in half and four or five were co-mounted on a single glass slide with double-sided tape (Fig. 1). For the rotated filters (Fig. 1A), the laser raster pattern comprised parallel lines that spanned the width and ran the length of the co-mounted half-filters. For the non-rotated filters (Fig. 1B), half-filters were ablated individually, except for stages 3 and 4, which were ablated spot-by-spot (5 of 10 spots in stage 3; 9 of 20 spots in stage 4). A standard reference material (NIST 2783, air particulate on filter media) with certified elemental areal mass densities (ng cm<sup>-2</sup>) and a lab blank (an unexposed filter, taken directly from the box) were mounted alongside the exposed filters. At least 1 cm<sup>2</sup> of the NIST and blank filters were ablated in four segments (0.25 cm<sup>2</sup> per segment), alternating with the sample filters. No loss in laser stability or ICP-MS drift was observed over 11 h.

The laser beam energy was sufficient to remove several layers of the PTFE filter in each particle size range, and no particles were visible on the filters (100× magnification) following ablation; hence, we assume that most particles were removed during ablation. However, deeply impacted particles may have remained embedded in the filters, particularly for the smaller particles in the asso of MOUDI sampling without rotation and high loading. Elemental concentrations were

20 the smaller particles in the case of MOUDI sampling without rotation and high loading. Elemental concentrations were determined via one-point calibration with the NIST standard.

#### 2.4 QA/QC

- Filters from sample 1 (stages 6–10) were cut in half and analysed by laser ablation and wet-chemical ICP-MS (Agilent Technologies 7900). For wet-chemical analysis, standards were prepared by diluting certified, traceable, inductively coupled plasma-grade, single-element standards. Filters were placed in metal-free HDPE vials containing 10 mL of an acid mixture (5 % HNO<sub>3</sub> and 2.5 % HCl, v/v), mixed by a rotary shaker for 12 h, and centrifuged. Extracts were measured without dilution. Recovery rates (85–105%) were measured using NIST SRM 2783 as a reference standard. Good agreement was
- 30 observed between wet chemical and laser ablation ICP-MS for As, Fe, Mn, Pb, V, and Zn (R = 0.94-0.99; m = 0.83-1.52). Poor agreement was observed for Cu, Ni, and Sb, attributed in part to low concentration levels in the extracts (low µg L<sup>-1</sup>) (Table 3).

#### **3** Results and discussion

#### 5 3.1 Elemental image maps

10

Unlike wet chemical ICP-MS (a bulk technique), LA ICP-MS allows for microanalysis of solid samples. In this work, each laser pulse provided spatially resolved elemental data (in counts per second associated with a certain m/z value). Using ImageJ software (Schneider et al., 2012), these data were mapped into pixels. The pixels formed elemental image maps with pixel size  $P_{size}$  (µm<sup>2</sup>):

$$P_{size} = ss \cdot t_{acg} \cdot d \tag{1}$$

where *ss* is the laser scanning speed ( $\mu$ m s<sup>-1</sup>),  $t_{acq}$  is the ICP-MS acquisition time (s), and *d* is the laser beam dimension ( $\mu$ m). 15 To obtain maps with square pixels we selected a scanning speed and an acquisition time so that  $ss \times t_{acq} = d$ ; hence,  $P_{size} = d^2$ . In this work,  $P_{size} = 150 \times 150 \ \mu\text{m}^2$  (*ss* = 300  $\mu\text{m}$  s<sup>-1</sup>,  $t_{acq} = 0.5$  s, *d* [square mask] = 150  $\mu$ m). For an analyzed area  $A_{anal}$  (cm<sup>2</sup>), the number of pixels *P* is given by  $10^8 \times A_{anal}/d^2$ .

For our half-filters,  $A_{anal}$  was nominally 2.4 cm<sup>2</sup> or 10,677 pixels (1 cm<sup>2</sup> = 4444 pixels). Image maps for Pb are shown in Figs. 1A (sample 1, rotated filters of stages 7–10) and 1B (sample 3, non-rotated filters of stages 6–9). (See Figs. S1 and S2 for maps of other elements in samples 1 and 3, respectively.) The false image map colors in Fig. 1 are brightness values, where each pixel *i* with element intensity  $I_i$  is converted by ImageJ to a brightness value  $B_i$  (dimensionless) using 65,536 pseudocolors (or 65,536 levels of gray).

#### 3.2 Visual inspection of elemental maps

Elemental image maps offer a robust tool for observing details about particle deposition not detectable by bulk methods. For example, the 2-D image maps in Fig. 1A illustrate the concentric circles created by MOUDI rotation. A clogged nozzle is apparent in stage 7, where minimal deposition is observed near the center of the half-filter. Other images made apparent a thumb print, a scissors cut, and the edge of the mounting tape. The ability to "see" such errors made it straightforward to avoid these parts of the filter when selecting areas to analyze. The maps also offered insights into the deposition patterns of various elements. Most notable was Ni, which unlike other elements, deposited in mounds even when collected with rotation

Additional information can be gleaned from 3-D image maps. For example, 3-D maps of non-rotated filters gave direct evidence for particle bounce (Marple et al., 2014). Relative intensities, measured with ImageJ, indicated that 12 % of the signal for Pb (stage 5) was located between spots (Fig. S4). Bounce was less pronounced in stages 6–10 (due to more

<sup>30 (</sup>Fig. S3).

nozzles). 3-D maps also made apparent "spikes" in the data, defined as values more than twice the median. Spikes were observed for most elements in sample filters (rotated), filter blanks, and gas blanks. In each case, outliers were replaced with the median of the pixels in the surrounding area ( $2 \times$  pixel size). This correction was made in all filters (including blanks) except for non-rotated filters, where the much higher concentrations in the spots masked the spikes. Spike removal is

5 illustrated in Fig. S5 for Zn (stages 6–10). After spike removal, the average relative standard deviation of the mapped area decreased from 300 to 30 %.

#### 3.3 Statistical analysis of elemental maps

The elemental brightness values associated with each pixel were analyzed statistically using ImageJ (after spike removal). First, we investigated how small an area could be ablated and still reproduce the mean half-filter value  $B_{half}$ . We measured

- 10 (for rotated filters) mean elemental areal brightness values,  $B_{areal}$  (cm<sup>-2</sup>) =  $\sum B_i / A_{anal}$ , for successively smaller areas  $A_{anal}$  (generally rectangular shapes, in different sections of the half-filter) and compared them to  $B_{half}$ . For stages 6–10, 10 % of the total filter deposit area (0.5 cm<sup>2</sup>, 20 min ablation time) gave brightness values in good agreement (±10 %) to the half-filter values. Representative results for Pb (sample 1, stages 7–9) are shown in Fig. 2. For stages 1–5, 20 % of the filter deposit area (1 cm<sup>2</sup>, 40 min ablation time) gave similar results. A notable exception was Ni, which had irregular deposition (Fig. S3).
- For non-rotated filters (without spike removal), good agreement ( $\pm 10$  %) to  $B_{half}$  was observed when (a) in stages 7–10, at least 15 % of the total deposit area was ablated and (b) in stages 5 and 6, areas containing 5 spots (of 40) and 8 spots (of 80), respectively, were ablated. In stages 3 and 4, where individual spots were ablated, we measured only the relative standard deviations across spots: 25 % (n = 5) and 23 % (n = 9), respectively.
- Second, for rotated filters, where deposition is expected to be uniform, we measured the relative standard deviations (*RSD*) of the elemental areal brightness values for each MOUDI stage (after spike removal). For stages 1 to 3, *RSD* values were large (120 to 80 %, respectively) due to fewer particles and fewer nozzles at those stages. *RSD* values decreased in stage 4 (60 %), stage 5 (40 %), and stages 6–10 (all ~30 %). *RSD* values became constant at larger areas (more pixels); hence, *RSE* values (= *RSD*/ $\sqrt{(number of pixels))}$  were also determined. Results are shown in Fig. 3, where both theoretical and experimental *RSE* values for Pb (sample 1) are plotted. For stage 4, an ablation area of 20 % (1 cm<sup>2</sup>, 40 min ablation time) was sufficient for good precision (*RSE* = 1 %); for stages 5–10, only 10 % was required (0.5 cm<sup>2</sup>, 20 min ablation time). In contrast to our previous findings (Gligorovski et al., 2008), in this work, the NIST standard also showed good precision for a 1 cm<sup>2</sup> (4444 pixels) ablation area (*RSD* = 22 %, *RSE* = 0.33 %).

#### **3.4 Elemental concentrations**

30

Like wet chemical ICP-MS, the ultimate goal of LA ICP-MS is to measure elemental concentrations. Elemental image maps were converted to concentrations using the NIST standard. For rotated filters,  $B_{areal}$  (cm<sup>-2</sup>) was converted to a mass density

 $M_{areal}$  (ng cm<sup>-2</sup>) using  $B_{areal,NIST}$  and  $M_{area,NIST}$  (eq 2).  $B_{areal}$  values were blank corrected using the average elemental areal brightness value of seven clean blanks  $B_{areal,BL}$ :

$$M_{areal} = \frac{B_{areal} - B_{areal,BL}}{B_{areal,NIST} / M_{areal,NIST}}$$
(2)

- Atmospheric elemental concentrations  $C_{air}$  (ng m<sup>-3</sup>) were determined by multiplying  $M_{areal}$  by the filter exposure area (4.91 cm<sup>2</sup>) and dividing by the air volume (43.2 m<sup>3</sup>). Although half-filter areas were used in these calculations, as shown above, 10 % areas (stages 6–10) or 20 % areas (stages 1–5) gave comparable results (± 10 %). A corresponding approach was used for non-rotated filters, except that spikes were not removed. Also, in stages 3 and 4, elemental brightness values were determined per spot rather than per area, then multiplied by the total number of spots per filter.
- Atmospheric elemental concentrations were highest in MOUDI stages 5–9 (1.0–0.1 μm); these concentrations are shown in Fig. 4A (Ljubljana, samples 1 and 2) and 4B (Martinska, samples 3 and 4). (See Tables S1 and S2 for all concentrations). Elemental concentrations (ng m<sup>-3</sup>) are shown on the left; percentages (normalized to 100 %) are shown on the right. To facilitate comparison, the two Ljubljana samples (24 and 72 h) and the two Martinksa samples (both 24 h) are plotted side by side. Together, these graphs illustrate both the magnitude and relative contributions of the nine elements in each stage.
- Several trends are worth noting. First, at both sites, the largest concentrations were observed in stage 5 (cutpoint = 1 μm). In Ljubljana, the 24 h concentrations were generally greater than the 72 h values (except for stage 5), but the relative percentages in each stage were quite similar. These trends suggest a common major source for the elements, but one that varies in magnitude from day to day. Consistent with previous works (Grgić et al., 2009; Hitzenberger et al., 2006; Mirage, 1989; Pacyna and Pacyna, 2001), we attribute this source to traffic emissions. Fe and Zn were the major elements comprising
- 85 % of the total elemental mass in sample 1 (101 ng m<sup>-3</sup> Fe; 44 ng m<sup>-3</sup> Zn) and 90 % of the total elemental mass in sample 2 (127 ng m<sup>-3</sup> Fe; 27 ng m<sup>-3</sup> Zn). The other trace elements (e.g., Cu, Pb, V, and Mn) were also consistent with traffic emissions and vehicle exhaust or fossil fuel or biomass combustion (Mirage, 1989; Pacyna and Pacyna, 2001). We note the excellent sensitivity that was observed in detecting trace metals with 24 h concentrations as low as 20 ( $\pm$  0.2), 22 ( $\pm$  0.2), and 26 ( $\pm$  0.1) pg m<sup>-3</sup> for V (stage 9), Mn (stage 10), and As (stage 10), respectively.
- In Martinska (Figure 4B), the total elemental concentrations were lower than in Ljubljana by roughly a factor of two. The highest values were in stage 5 (36 ng m<sup>-3</sup>), predominated by Fe (92 %) with smaller amounts of Zn (3 %), Mn (2 %), Pb (1 %), and V (1 %). In general, higher total concentrations were observed in sample 4 (day 2), and there was more variability in composition between the two days than in Ljubljana. The largest variability was observed for V in stage 8; concentrations varied from 1.92 (39 %) in sample 3 (day 1) to 1.36 ng m<sup>-3</sup> (14 %) in sample 4 (day 2). Vanadium has been
- observed previously in marine aerosols (Turšič et al., 2006) and is attributed to continental pollution from oil combustion (Tolocka et al., 2004). The variability in the direction of continental winds on the two days of sampling may have influenced this signal.

# 5 4 Conclusions

In this proof of concept paper, we have demonstrated the usefulness of LA ICP-MS as a tool for analysing the elemental composition of size-segregated atmospheric particles collected on filter-based media. Previous problems associated with LA ICP-MS were addressed: (1) MOUDI rotation sampling overcomes the lack of uniformity in particle deposition, creating a sample highly suitable for LA-ICP-MS 2D mapping, 2) the 2D mapping mode yields results which show a high degree of 10 accuracy when larger areas are ablated and superior detection limits, and 3) quantification problems due to non-matrix matched standards are circumvented by ablating through the filter or obliterating the particles on the filters, warranting the reliable use of one-point calibrating on NIST SRM 2783. Together, these improvements allowed for an efficient and sensitive measurement of elemental composition. Although half-filters were analysed in much of this work, we showed that comparable results could be obtained by ablating only 1  $\text{cm}^2$  of filter or less. The ability to analyse a filter in roughly 40 min 15 of instrument time makes feasible routine measurements of size-segregated particles. Compositional graphs of these particles, such as those shown in Fig. 4 for Ljubljana and Martinska, will be useful to the atmospheric community by allowing comparison of elemental profiles of particulate collected at diverse sites (e.g., urban industrial centres to remote background locations). Such profiles can be compared over days, months, or even years; short-term and long-term compositional changes can be used to monitor atmospheric changes such as a new pollution source, the impacts of pollution 20 remediation, and the effects of climate change. A key limitation to this approach is the lack of a size-segregated reference standard; hence, measurements of absolute elemental concentrations is not vet feasible. Nonetheless, much can be learned

from relative changes in elemental composition, which are easily measured by this technique.

- Supplementary material. Supplementary materials include (1) atmospheric elemental concentrations for MOUDI stages 1–5 (Table S1) and 6–10 (Table S2); (2) elemental image maps for stages 6–10 of sample 1 (Fig. S1), (3) elemental image maps for stages 6 and 9 of sample 2 ("spots") (Fig. S2), (4) elemental image maps of Ni vs Pb (Fig. S3, stages 3–8), (5) illustration of particle bounce for Pb (Fig. S4), and (6) illustration of spike removal for zinc (Fig. S5, stages 6–10, sample 1).
- 30 Author contribution. M. S. Robinson collected samples, carried out experiments, and prepared the manuscript. V. S. Šelih and M. Šala carried out experiments. I. Grgić collected samples and prepared the manuscript. M Bitsui collected samples. J. T. van Elteren designed and carried out experiments and prepared the manuscript.

*Acknowledgements*. This work was funded by the Slovenian Research Agency (Contract No. P1-0034) and the Fulbright Scholar Program, sponsored by the U.S. Department of State, administered by CIES, a division of IIE. The authors thank Dr. Sanja Frka Milosavljević for arranging access to Martinska station.

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

Table 1. Optimized operating conditions for the laser ablation ICP-MS system.

| Laser ablation (Analyte G2)        |                                                                                                                                 |
|------------------------------------|---------------------------------------------------------------------------------------------------------------------------------|
| Wavelength                         | 193                                                                                                                             |
| Pulse length                       | <4 ns                                                                                                                           |
| Beam size (square mask)            | 150 μm                                                                                                                          |
| Fluence                            | $1.21 \text{ J cm}^{-2}$                                                                                                        |
| Repetition rate                    | 10 Hz                                                                                                                           |
| Scanning speed                     | $300 \ \mu m \ s^{-1}$                                                                                                          |
| He carrier flow rate               | $0.5 \text{ Lmin}^{-1}$ (cup) and $0.3 \text{ Lmin}^{-1}$ (cell)                                                                |
| Ar make-up flow rate               | $0.8 \text{ L min}^{-1}$                                                                                                        |
|                                    |                                                                                                                                 |
| ICP-MS (Agilent 7900)              |                                                                                                                                 |
| Rf power                           | 1500 W                                                                                                                          |
| Sampling depth                     | 9 mm                                                                                                                            |
| Acquisition time/mass              | 0.5–1.0 s                                                                                                                       |
| Measurement mode                   | time-resolved TRA(1)                                                                                                            |
| Ar plasma gas flow rate            | $15 \mathrm{L min}^{-1}$                                                                                                        |
| Ar auxiliary gas flow rate         | $0.7 L \min^{-1}$                                                                                                               |
| No. of line scans/mapping sequence | $70-150 (0.56-4.38 \text{ min line scan}^{-1})$                                                                                 |
| Isotopes measured                  | <sup>75</sup> As, <sup>63</sup> Cu, <sup>57</sup> Fe, <sup>55</sup> Mn, <sup>60</sup> Ni, <sup>208</sup> Pb, <sup>121</sup> Sb, |
|                                    | <sup>51</sup> V, <sup>66</sup> Zn                                                                                               |

**Table 2**. Detection limits DL (3 x SD) were determined from the mean standard deviations of seven clean blanks with ablation areas of  $1.1 \text{ cm}^2$ . DLs were converted to concentrations using the NIST standard and a theoretical air sampling period of 24 h at 30 LPM. All values were spike corrected. Units are in ng m<sup>-3</sup>.

| element | DL    |  |
|---------|-------|--|
| As      | 0.019 |  |
| Cu      | 0.009 |  |
| Fe      | 0.370 |  |
| Mn      | 0.016 |  |
| Ni      | 0.114 |  |
| Pb      | 0.004 |  |
| Sb      | 0.019 |  |
| V       | 0.003 |  |
| Zn      | 0.160 |  |

**Table 3.** Correlation coefficient (*R*) and slope (*m* in y = mx) for concentrations measured by laser ablation ICP-MS (x) and wet-chemical ICP-MS (y) in MOUDI stages 6–10. Reasonable agreement was observed for the first six elements.

| element | R    | т    |  |
|---------|------|------|--|
| As      | 0.97 | 0.92 |  |
| Fe      | 0.99 | 1.52 |  |
| Mn      | 0.97 | 0.93 |  |
| Pb      | 0.98 | 0.91 |  |
| V       | 0.99 | 0.83 |  |
| Zn      | 0.95 | 0.92 |  |
| Cu      | 0.78 | 1.08 |  |
| Ni      | 0.03 | 2.20 |  |
| Sb      | 0.52 | 0.23 |  |

# Figure captions.

5

15

**Figure 1.** Setups for LA ICP-MS of rotated (A, sample 1, stages 7–10) and non-rotated (B, sample 3, stages 6–9) filters; both optical and elemental images (pseudocolored image maps of Pb) for each filter are shown. Lab blanks (LB) and the NIST 2783 standard (STD) were analysed next to the filters.

**Figure 2.** Comparison of mean brightness values for smaller ablation areas (Bfraction) to the half-filter value (Bhalf) for Pb in sample 1 (stages 7–9). The blue bar shows that ablation of smaller areas agreed with the half-filter value to  $\pm 10$  %.

Figure 3. Relative standard errors ( $RSE = RSD/\sqrt{P}$ ) as a fraction of the number of pixels *P* (=21,822 pixels, associated with a total filter deposit area  $A_{tot}$  of 4.91 cm<sup>2</sup>). Solid lines are theoretical values for RSD = 30, 40, and 60 %. Markers are experimental values for Pb in stages 4 (filled squares), 5 (filled triangles), and 7 and 8 (open squares and circles).

**Figure 4.** Elemental concentrations of (A) samples 1 (24 h) and 2 (72 h) in Ljubljana and (B) samples 3 (24 h) and 4 (24 h) in Martinska. Graphs on the left show atmospheric concentrations. Graphs on the right show corresponding percent composition. MOUDI stages 5–9 correspond to cutpoint dia. of 1.0, 0.56, 0.32, 0.18, and 0.10 µm, respectively.

FIG2