# Peer review of "Laser Ablation ICP-MS of Size-Segregated Atmospheric Particles Collected with a MOUDI Cascade Impactor: A Proof of Concept"

_Atmospheric Measurement Techniques, 2016_

## Referee Comment (RC1) · Anonymous Referee #2 · 27 Feb 2017

The paper dedicated to the analysis by LA ICP-MS of atmospheric particles collected with a MOUDI cascade impactor fit well with the scope of the AMT journal. However, many questions arise. The study which intended to demonstrate the value of this combination (MOUDI / LA ICP-MS) not treated and discussed a lot of keypoints that deserved the proof of concept.

The main problem comes with the lack of a real reference material characterized for the different particle sizes tested here. The urban particulate matter of the SRM 2783 used as reference in this study has a median particle size of 3.2 $\mu$m and a size range of 2.5 $\mu$m and particles are collected on nuclepore polycarbonate membrane filters of 0.4 $\mu$m pore size. How it can be possible for the authors to compare its response by LA

ICPMS with what they obtained on their particulate samples of lower sizes, collected on PTFE filters? This is not discussed (influence of the particle size and the filter type on the laser ablation efficiency / influence of the laser beam mask size on the laser ablation efficiency and the mapping...)

The authors claimed that "The laser beam energy was sufficient to remove all of the particles during ablation, allowing elemental concentrations to be determined via a one-point calibration with the NIST standard" (P3L12). The authors have to prove that it is realistic. What's about deep-impacted particles into filters? As they mentioned in their previous paper published in Science of the Total Environment in 2008 ("A multi-element mapping approach for size-segregated atmospheric particles using laser ablation ICP-MS combined with image analysis") distribution of particulate matter in SRM 2783 is non-homogeneous... Why is it not taking into account here?

LA ICP-MS and "wet chemical" ICP-MS of several samples had been also compared for 5 samples which is informative but non-sufficient to understand the effect of the particulate size on the LA ICP-MS response.

Particulate collection is another keypoint of this particulate matter analysis. However, collection particle losses during MOUDI collection was not investigated (nozzle wall loss which is dependent on the size but also on the particle composition / clogging effects). Moreover, the effect of the rotation of the filter was studied by comparing the results got from two different cities, at different days and different climatic conditions...too much parameters varied to get really confident comparison.

Moreover, no validation data are given on this new measurement system. Validation steps of the concept should be considered (repeatability integrating the collection step).

Why no conclusion was drawn on the proof of concept...?

I suggest conducting the proof of concept on a very well characterized particulate sample (size/composition by other analytical tools), to study systematically each critical parameter and compare with the author's previous work and the literature (one important paper in this field is missed: Hsieh, Chen et al. "Elemental analysis of airborne particulate matter using an electrical low-pressure impactor and laser ablation/inductively coupled plasma mass spectrometry" J.Anal.At.Spectrom., 2011, 26, 1502).

---

## Referee Comment (RC2) · Anonymous Referee #1 · 3 Mar 2017

The paper investigates " Laser Ablation ICP-MS of Size-Segregated Atmospheric Particles Collected with a MOUDI Cascade Impactor: A Proof of Concept" The topic of the manuscript is very interesting and not yet investigated in literature to the best of my knowledge. The manuscript is concise and well written and conclusion adequately supported by experimental data. I suggest publication in Atmospheric Measurement techniques journal pending minor revision as noted: The developed method is quite promising for analysis of elemental composition of size-segregated atmospheric particles collected on filters. The authors compared this method with the "wet chemical" ICP-MS. However, a comparison with other techniques from the literature to validate their method is missing. For example, how this technique proofs useful compared

with some instruments aimed for online analysis of the elemental composition in single atmospheric particles, such as A-TOF-MS, for instance. A-TOF-MS is also based on laser desorption technique. Furthermore, the authors should better explain what are really the advantages and disadvantages of their method. Therefore, a discussion about atmospheric implications of this method should improve the quality of the manuscript and make it more interesting for the readers. I suggest including a section "Atmospheric implications" How was chosen the NIST standard? I this the best option for this kind of study? On which scientific basis were chosen the 9 elements in this study? On which basis the authors decided to cut the filters? How do we know for sure that the elements are homogenously distributed on the filters? I did not understand why some filters are ablated individually and some spot by spot? How was this decided?

---

## Author Response (AR1)

[revised manuscript text omitted]

**Figure captions.**

**Figure 1.** Setups for LA ICP-MS of rotated (A, sample 1, stages 7–10) and non-rotated (B, sample 3, stages 6–9) filters; both optical and elemental images (pseudocolored image maps of Pb) for each filter are shown. Lab blanks (LB) and the NIST 2783 standard (STD) were analysed next to the filters.

**Figure 2.** Comparison of mean brightness values for smaller ablation areas (Bfraction) to the half-filter value (Bhalf) for Pb in sample 1 (stages 7–9). The blue bar shows that ablation of smaller areas agreed with the half-filter value to ±10 %.

**Figure 3.** Relative standard errors ($RSE = RSD/\sqrt{P}$) as a fraction of the number of pixels $P$ (=21,822 pixels, associated with a total filter deposit area $A_{tot}$ of 4.91 cm$^2$). Solid lines are theoretical values for $RSD$ = 30, 40, and 60 %. Markers are experimental values for Pb in stages 4 (filled squares), 5 (filled triangles), and 7 and 8 (open squares and circles).

**Figure 4.** Elemental concentrations of (A) samples 1 (24 h) and 2 (72 h) in Ljubljana and (B) samples 3 (24 h) and 4 (24 h) in Martinska. Graphs on the left show atmospheric concentrations. Graphs on the right show corresponding percent composition. MOUDI stages 5–9 correspond to cutpoint dia. of 1.0, 0.56, 0.32, 0.18, and 0.10 µm, respectively.

FIG1

[Figure]

FIG2

[Figure]

**FIG3**

[Figure]

**FIG4**

[Figure]

(A) Ljubljana (samples 1 and 2)

[Figure]

(B) Martinska (samples 3 and 4)

[Figure]

**Reply to reviewers (original manuscript):**

**Responses to Reviewer #1**

**(1)** *Reviewer's comment: This manuscript was previously submitted to another journals with other scope requirements different than the scope of AMT. In order to render the manuscript suitable for the AMT discussion some changes should be performed. For example, the conclusion section is completely missing.*

**Authors' response:** This paper was formatted initially for a journal with stricter page limitations; hence, much information was moved to Supplementary Materials. We have now moved three tables back into the manuscript (Tables 1−3) and added a Conclusions section. We also included two more samples (a 72 h sample from Ljubljana and a second 24 h sample from Martinska). The results of all four samples are now discussed and shared in a new figure (Fig. 4).

**Authors' changes in manuscript:** Addition of Tables 1−3 to manuscript; addition of Fig. 4 (which describes results of four MOUDI samples rather than two); addition of a Conclusions section.

**(2)** *Reviewer's comment: Comparison with other similar techniques to validate their method is also missing.*

**Authors' response:** Based on the concentration levels found in the filters, validation with similar imaging techniques like XRF, PIXE, etc. is questionable as LA-ICP-MS is the most sensitive technique for elemental microanalysis. However, by comparing the elemental concentration in size-segregated particles with a sensitive bulk analytical technique like ICP-MS, after digestion of the filter, an indirect comparison can be made as explained in section 2.4 and highlighted in Table 3. As such we feel that the LA-ICP-MS imaging method used yields accurate and precise data.

**(3)** *Reviewer's comment: The authors never mentioned what are the advantages and disadvantages of this method. Therefore, a discussion about atmospheric implications of this method should improve the quality of the manuscript and make it more interesting for the readers*

**Authors' response:** We agree with the reviewer and made changes to the text.

**Authors' changes to manuscript:** The Conclusions section now highlights the strengths and weaknesses of the approach and describe its potential usefulness to the atmospheric community.

**(4)** *Reviewer's comment: There are many tables and figures in the supplementary section while the manuscript on the other hand is way too short. I would recombine the manuscript and the supplementary material when possible.*

**Authors' response:** As noted in comment #1, we have now integrated Tables 1–3 into the manuscript.

**Authors' changes to manuscript:** Addition of Tables 1–3.

**Responses to Reviewer #2**

**(1)** *Reviewer's comment: The manuscript reports the analysis by LA ICP-MS of atmospheric particles collected with a MOUDI cascade impactor. This subject fit well with the scope of the AMT journal. However, this study which intended to demonstrate the value of this combination not treated and discussed a lot of keypoints that deserved the proof of concept:*

*The main problem comes with the lack of a real reference material characterized for the different particle sizes tested here. The urban particulate matter of the SRM 2783 used as reference in this study has a median particle size of 3.2 µm and a size range of 2.5 µm and particles are collected on nuclepore polycarbonate membrane filters of 0.4 µm pore size. How it can be possible for the author to compare its response by LA ICPMS with what they obtained on their particulate samples of lower sizes, collected on PTFE filters? This is not discussed (influence of the particle size and the filter type on the laser ablation efficiency/influence of the laser beam mask size on the ablation efficiency and the mapping…)*

**Authors' response:** We used NIST SRM 2783 as an absolute calibration standard, i.e. the LA-ICP-MS parameters were such that the laser beam completely penetrated the nucleopore filter, making matrix-matching unnecessary. The PTFE filters were not completely obliterated after ablation, probably due to the fact they were ticker, but the size-segregated particles on the filter were in most instances completely removed as observed by optical microscopy under 100× magnification, except maybe for the deepest impacted, smallest particles in the case of MOUDI sampling without rotation, making quantification by one-point calibration of the rotational filters on the NIST standard accurate. Since we use a laser beam diameter of 150 µm with a square mask it is obvious that the particles ablated are much smaller and as such the influence of particle size on the quantification is negligible.

**Authors' changes to manuscript:** We state that quantification is related to the use of NIST SRM 2783 as an absolute calibration standard and as such matrix-matching becomes unnecessary.

(2)  **Reviewer's comment:** *The authors claimed that "The laser beam energy was sufficient to remove all of the particles during ablation, allowing elemental concentrations to be determined via a one-point calibration with the NIST standard" (P3L12). The authors have to prove that it is realistic. What about deep-impacted particles into filters?*

**Authors' response:**  Visual inspection of the post-ablation filters under 100× magnification showed that the selected laser parameters (e.g., fluence/mask size/repetition rate) were sufficient to remove multiple layers of PTFE in each particle stage; hence, we assumed that the particles themselves were also ablated. However, it is possible that the deepest impacted particles were missed, especially in the smallest size ranges in the case MOUDI without rotation and high loading of particles. MOUDI with rotation prevents such potential problems altogether.

**Authors' changes to manuscript:** In Section 2.3 (Filter preparation and ablation), we now point out that although no particles were visible on the filters after ablation, that it is possible that small, deeply impacted particles may have remained embedded in the filters.

(3)  **Reviewer's comment:** *As they mentioned in their previous paper published in Science of the Total Environment in 2008 ("A multi-element mapping approach for size-segregated atmospheric particles using laser ablation ICP-MS combined with image analysis") distribution of particulate matter in SRM 2783 is non-homogeneous.*

**Authors' response:**  We are aware of the heterogeneity of the NIST standard as the certificate declares that a sampling area of 1 cm$^2$ is deemed necessary for reaching the certified uncertainty. In the current manuscript we routinely analyzed 1 cm$^2$ to comply with these requirements.

**Authors' changes to manuscript**: In section 3.3, we have now added the following: In contrast to our previous findings (Gligorovski et al., 2008), in this work, the NIST standard gave good precision for a 1 cm$^2$ (4444 pixels) ablation area ($RSD = 22$ %, $RSE = 0.33$ %).

(4)  **Reviewer's comment:** *LA ICP-MS and "wet chemical" ICP-MS of several samples had been also compared for 5 samples which is informative but non-sufficient to understand the effect of the particulate size on the LA ICP-MS response.*

**Authors' response:**  As stated in comment #1, we did not specifically examine the effect of particulate size on the LA ICP-MS response; however, we did optimize the laser parameters so that particles in all stages appeared to be ablated. Furthermore, section 2.4 (QA/QC) clearly shows that there is a good agreement between LA-ICP-MS and bulk ICP-

MS after digestion for most elements and for stages 6-10 with cut-point dia. of 0.56, 0.32, 0.18, 0.10, and 0.0156 μm, respectively.

**Authors' changes to manuscript:** None.

(5) ***Reviewer's comment:*** *Particulate collection is another keypoint of this particulate matter analysis. However, collection particle losses during MOUDI collection was not investigated (nozzle wall loss which is dependent on the size but also on the particle composition / clogging effects). Moreover, the effect of the rotation of the filter was studied by comparing the results got from two different cities, at different days and different climatic conditions…too much parameters varied to get really confident comparison.*

**Authors' response:** It is true that there may have been particle losses during MOUDI collection, but this was not our focus. Our focus was to use LA ICP-MS (as an alternative to wet-chemical ICP-MS) to determine the elemental composition of the particles that were successfully collected. We also used laser ablation to investigate the uniformity of the particles that were deposited when the MOUDI was rotated.

**Authors' changes to manuscript:** None.

(6) ***Reviewer's comment:*** *No conclusion was drawn on the proof of concept…*

**Authors' response:** We have now added a Conclusions section.

**Authors' changes to manuscript:** We have added a Conclusions section.

(7) ***Reviewer's comment:*** *I suggest conducting the proof of concept on a very well characterized particulate sample (size/composition by other analytical tools), to study systematically each critical parameter and compare with their previous work and the literature (one important paper in this field is missed: Hsieh, Chen et al. "Elemental analysis of airborne particulate matter using an electrical low-pressure impactor and laser ablation/inductively coupled plasma mass spectrometry" J.Anal.At.Spectrom., 2011, 26, 1502). Validation steps of the concept need also to be considered (repeatability integrating the collection step). Consequently, I recommend that authors go further in their analytical study before to be resubmitted for review in this journal.*

**Authors' response:** The current manuscript, although it does not study systematically each critical parameter, shows that LA-ICP-MS yields accurate data and superior detection limits for elemental analysis of size-segregated aerosols

collected by MOUDI in rotation mode. Please also see the response to comments # 1 and 4. We are confident that this progress will be of interest to the atmospheric community.

**Authors' changes to manuscript:** The reference mentioned was included in the manuscript. The importance of our findings is more clearly stated in the Conclusions section, which was lacking in the first submission.

**Interactive comments and authors' response I:**

*The paper dedicated to the analysis by LA ICP-MS of atmospheric particles collected with a MOUDI cascade impactor fit well with the scope of the AMT journal. However, many questions arise. The study which intended to demonstrate the value of this combination (MOUDI / LA ICP-MS) not treated and discussed a lot of keypoints that deserved the proof of concept. The main problem comes with the lack of a real reference material characterized for the different particle sizes tested here. The urban particulate matter of the SRM 2783 used as reference in this study has a median particle size of 3.2 μm and a size range of 2.5 μm and particles are collected on nuclepore polycarbonate membrane filters of 0.4 μm pore size. How it can be possible for the authors to compare its response by LA ICPMS with what they obtained on their particulate samples of lower sizes, collected on PTFE filters? This is not discussed (influence of the particle size and the filter type on the laser ablation efficiency / influence of the laser beam mask size on the laser ablation efficiency and the mapping...)*

**Authors' response:** We used NIST SRM 2783 as an absolute calibration standard, i.e. the LA ICP-MS parameters were such that the laser beam completely penetrated the nucleopore filter, making matrix-matching unnecessary. The PTFE filters were not completely obliterated after ablation, probably due to the fact they were thicker, but the size-segregated particles on the filter were in most instances completely removed as observed by optical microscopy under 100× magnification, except maybe for the deepest impacted, smallest particles in the case of MOUDI sampling without rotation, making quantification by one-point calibration of the rotational filters on the NIST standard accurate. Since we use a laser beam diameter of 150 μm with a square mask it is obvious that the particles ablated are much smaller and as such the influence of particle size on the quantification is negligible.

*The authors claimed that "The laser beam energy was sufficient to remove all of the particles during ablation, allowing elemental concentrations to be determined via a one-point calibration with the NIST standard" (P3L12). The authors have to prove that it is realistic. What's about deep-impacted particles into filters? As they mentioned in their previous paper published in Science of the Total Environment in 2008 ("A multi-element mapping approach for size-segregated atmospheric particles using laser ablation ICP-MS combined with image analysis") distribution of particulate matter in SRM 2783 is non-homogeneous... Why is it not taking into account here?*

**Authors' response:** Visual inspection of the post-ablation filters under 100× magnification showed that the selected laser parameters (e.g., fluence/mask size/repetition rate) were sufficient to remove multiple layers of PTFE in each particle stage; hence, we assumed that the particles themselves were also ablated. However, it is possible that the deepest impacted particles were missed, especially in the smallest size ranges in the case of MOUDI sampling without rotation and high loading of particles. MOUDI with rotation prevents such potential problems altogether. We are aware

of the heterogeneity of the NIST standard as the certificate declares that a sampling area of 1 cm$^2$ is deemed necessary for reaching the certified uncertainty. In the current manuscript we routinely analyzed 1 cm$^2$ to comply with these requirements.

*LA ICP-MS and "wet chemical" ICP-MS of several samples had been also compared for 5 samples which is informative but non-sufficient to understand the effect of the particulate size on the LA ICP-MS response.*

**Authors' response:** We did not specifically examine the effect of particulate size on the LA ICP-MS response; however, we did optimize the laser parameters so that particles in all stages appeared to be ablated. Furthermore, section 2.4 (QA/QC) clearly shows that there is a good agreement between LA CP-MS and bulk ICP-MS after digestion for most elements and for stages 6-10 with cut-point dia. of 0.56, 0.32, 0.18, 0.10, and 0.0156 μm, respectively.

*Particulate collection is another keypoint of this particulate matter analysis. However, collection particle losses during MOUDI collection was not investigated (nozzle wall loss which is dependent on the size but also on the particle composition / clogging effects). Moreover, the effect of the rotation of the filter was studied by comparing the results got from two different cities, at different days and different climatic conditions...too much parameters varied to get really confident comparison.*

**Authors' response:** It is true that there may have been particle losses during MOUDI collection, but this was not our focus. Our focus was to use LA ICP-MS (as an alternative to wet-chemical ICP-MS) to determine the elemental composition of the particles that were successfully collected. We also used laser ablation to investigate the uniformity of the particles that were deposited when the MOUDI was rotated.

*Moreover, no validation data are given on this new measurement system. Validation steps of the concept should be considered (repeatability integrating the collection step).*

**Authors' response:** Based on the concentration levels found in the filters, validation with similar imaging techniques like XRF, PIXE, etc. is questionable as LA ICP-MS is the most sensitive technique for elemental microanalysis. However, by comparing the elemental concentration in size-segregated particles with a sensitive bulk analytical technique like ICP-MS, after digestion of the filter, an indirect comparison can be made as explained in section 2.4 and highlighted in Table 3. As such we feel that the LA ICP-MS imaging method used yields accurate and precise data.

*Why no conclusion was drawn on the proof of concept ...?*

**Authors' response:**  The current manuscript contains a "Conclusions" section.

*I suggest conducting the proof of concept on a very well characterized particulate sample (size/composition by other analytical tools), to study systematically each critical parameter and compare with the author's previous work and the literature (one important paper in this field is missed: Hsieh, Chen et al. "Elemental analysis of airborne particulate matter using an electrical low-pressure impactor and laser ablation/inductively coupled plasma mass spectrometry" J.Anal.At.Spectrom., 2011, 26, 1502).*

**Authors' response:** The current manuscript, although it does not study systematically each critical parameter, shows that LA ICP-MS yields accurate data and superior detection limits for elemental analysis of size-segregated aerosols collected by MOUDI in rotation mode. The mentioned reference is included in the current manuscript.

**Interactive comments and authors' response II:**

*The paper investigates " Laser Ablation ICP-MS of Size-Segregated Atmospheric Particles Collected with a MOUDI Cascade Impactor: A Proof of Concept" The topic of the manuscript is very interesting and not yet investigated in literature*
5    *to the best of my knowledge. The manuscript is concise and well written and conclusion adequately supported by experimental data. I suggest publication in Atmospheric Measurement techniques journal pending minor revision as noted: The developed method is quite promising for analysis of elemental composition of size-segregated atmospheric particles collected on filters. The authors compared this method with the "wet chemical" ICP-MS. However, a comparison with other techniques from the literature to validate their method is missing. For example, how this technique proofs useful compared*
10    *with some instruments aimed for online analysis of the elemental composition in single atmospheric particles, such as A-TOF-MS, for instance. A-TOF-MS is also based on laser desorption technique*

**Authors' response**: Based on the concentration levels found in the filters, validation with similar imaging techniques like XRF, PIXE, etc. is questionable as LA ICP-MS is the most sensitive technique for elemental microanalysis. However, by
15    comparing the elemental concentration in size-segregated particles with a sensitive bulk analytical technique like ICP-MS, after digestion of the filter, an indirect comparison can be made as explained in section 2.4 and highlighted in Table 3. As such we feel that the LA ICP-MS imaging method used yields accurate and precise data. An instrument like A-TOF-MS is meant for the determination of single particle size and analysis individual particles and refractory materials such as sodium chloride, elemental carbon and mineral dust constituents. As such specific classes of particles are identified based on
20    fingerprinting or combing peaks but true elemental composition analysis as achieved with our approach is not possible. Although we limited ourselves to a limited suite of elements, in theory most element of the periodic table can be measured routinely.

*Furthermore, the authors should better explain what are really the advantages and disadvantages of their method.*
25    *Therefore, a discussion about atmospheric implications of this method should improve the quality of the manuscript and make it more interesting for the readers. I suggest including a section "Atmospheric implications".*

**Authors' response**: In this proof of concept paper, we have demonstrated the usefulness of LA ICP-MS as a tool for analysing the elemental composition of size-segregated atmospheric particles collected on filter-based media and have
30    addressed these issues in the Conclusions section.

Previous problems associated with LA ICP-MS were addressed: (1) MOUDI rotation sampling overcomes the lack of uniformity in particle deposition, creating a sample highly suitable for LA-ICP-MS 2D mapping, 2) the 2D mapping mode yields results which show a high degree of accuracy when larger areas are ablated and superior detection limits, and 3)

quantification problems due to non-matrix matched standards are circumvented by ablating through the filter or obliterating the particles on the filters, warranting the reliable use of one-point calibrating on NIST SRM 2783. Together, these improvements allowed for an efficient and sensitive measurement of elemental composition. Compositional graphs of particles such as those shown in Fig. 4 for Ljubljana and Martinska, will be useful to the atmospheric community by allowing comparison of elemental profiles of particulate collected at diverse sites (e.g., urban industrial centres to remote background locations). Such profiles can be compared over days, months, or even years; short-term and long-term compositional changes can be used to monitor atmospheric changes such as a new pollution source, the impacts of pollution remediation, and the effects of climate change.

*How was chosen the NIST standard? I this the best option for this kind of study? On which scientific basis were chosen the 9 elements in this study? On which basis the authors decided to cut the filters? How do we know for sure that the elements are homogenously distributed on the filters? I did not understand why some filters are ablated individually and some spot by spot? How was this decided?*

**Authors' response**: We used NIST SRM 2783 as an absolute calibration standard as this seems to be the only "reliable" elements standard available for particulate matter although we are aware of the heterogeneity of the NIST standard as the certificate declares that a sampling area of 1 cm$^2$ is deemed necessary for reaching the certified uncertainty. In the current manuscript we routinely analyzed 1 cm$^2$ to comply with these requirements. We have chosen the most important trace elements based on inhalation risks associated with particle-bound metals as explained in the introduction.

Filters were not cut but we measured half-filters although measurement of areas of 1 cm$^2$ would suffice for accurate analytical results. Since the laser samples a statistically significant portion of the filter we can be confident that heterogeneity issues are circumvented. Using the MOUDI with and without rotation, generating homogeneously distributed particles and "particle spots", respectively, we show two approaches used for field collection of size-segregated particles and the limitations of both approaches, i.e. higher sensitivity and noisier data in the case of "particle spots" and lower sensitivity and better reproducibility in the case of homogeneously distributed particles.